# Childhood Mandatory Vaccinations: Current Situation in European Countries and Changes Occurred from 2014 to 2024

**DOI:** 10.3390/vaccines12111296

**Published:** 2024-11-20

**Authors:** Sara Farina, Alessandra Maio, Maria Rosaria Gualano, Walter Ricciardi, Leonardo Villani

**Affiliations:** 1Section of Hygiene, University Department of Life Sciences and Public Health, Università Cattolica del Sacro Cuore, 00168 Rome, Italy; sara.farina02@icatt.it (S.F.); alessandra.maio02@icatt.it (A.M.); walter.ricciardi@unicatt.it (W.R.); 2Faculty of Medicine, UniCamillus—Saint Camillus International University of Health and Medical Sciences, 00131 Roma, Italy; mar.guala@gmail.com; 3Leadership Research Center, Università Cattolica del Sacro Cuore, Campus di Roma, 00168 Rome, Italy; 4Center for Global Health Research and Studies, Università Cattolica del Sacro Cuore, Largo Francesco Vito 1, 00168 Rome, Italy

**Keywords:** vaccination, mandatory vaccination, pediatric, law, public health

## Abstract

**Background/Objectives**: Vaccination is one of the most effective public health interventions, preventing millions of deaths globally each year. However, vaccine hesitancy, driven by misinformation and reduced disease risk perception, has led to declining vaccination rates and the resurgence of vaccine-preventable diseases (VPDs) in Europe. In response to this, countries have implemented various strategies, including mandatory and recommended vaccination programs. The objective of this study is to map the current European landscape of pediatric vaccination policies, and the variations that have occurred in the last decade. **Methods**: This rapid review was conducted on PubMed, Google, and the European Centre for Disease Prevention and Control website, to collect all vaccination schedules in EU/EEA countries in 2024 and all documents focusing on the introduction of mandatory vaccines during the last decade. **Results**: As of 2024, 13 countries had at least one mandatory pediatric vaccination, with France, Hungary, and Latvia requiring all but one vaccine. In contrast, 17 countries had no mandatory vaccinations, relying only on recommendations. Between 2014 and 2024, six countries (Croatia, France, Germany, Hungary, Italy, and Poland) introduced or extended mandatory vaccinations. **Conclusions**: European vaccination policies show significant variation. Effective programs depend on robust healthcare systems, public trust, and adaptable strategies to address vaccine hesitancy and the resurgence of VPDs.

## 1. Introduction

Vaccinations are universally recognized as one of the most effective public health interventions, significantly reducing the incidence, severity, and economic burden of numerous infectious diseases. For instance, the World Health Organization (WHO) estimates that vaccination prevents approximately 3.5–5 million deaths each year from diseases such as diphtheria, tetanus, pertussis, influenza, and measles [1]. Despite these successes, vaccine hesitancy, characterized by delays or refusals of vaccines despite their availability, has emerged as a growing concern in recent years [2]. Hesitancy is driven by many social, health, economic, and cultural factors, including misinformation, declining trust in healthcare systems, and a misperception of diminished disease risk due to the lower prevalence of infectious diseases [3,4,5]. Poor health literacy and the rapid dissemination of false information through the media further exacerbate the problem, leading to reduced vaccination coverage and the resurgence of vaccine-preventable diseases (VPDs) [6,7,8,9].

A notable example of this trend occurred in 2014–2015, when a global increase in measles cases, particularly in Europe, was linked to declining vaccination rates [10]. This prompted many nations to revise their vaccination policies, with the WHO’s European Regional Office calling for renewed efforts to boost measles and rubella vaccination, especially in high-risk groups [11]. The COVID-19 pandemic, which began in 2020, further highlighted the critical role of vaccination in managing public health crises, driving the need for flexible and robust vaccination strategies [12,13,14].

The resurgence of VPDs poses significant risks to vulnerable populations, including infants, the elderly, and immunocompromised individuals. Declining vaccination rates have also contributed to the rise of other serious infections, such as meningitis, influenza, and pneumonia, leading to increased hospitalizations and deaths [15].

In response to these challenges, European countries have adopted diverse strategies to improve vaccine coverage. Some have focused on public education and accessibility, while others have strengthened mandatory vaccination laws [16]. This study aims to map the landscape of mandatory and recommended pediatric vaccinations across Europe, analyzing how these policies have evolved in the last ten years. By examining historical trends and policy changes, the study seeks to provide insights for developing more effective vaccination strategies to address current and future public health challenges.

## 2. Materials and Methods

This rapid review aimed to map the current state of pediatric vaccination policies across European Union (EU) and European Economic Area (EEA) countries in the last decade, with a focus on whether vaccines have been made mandatory or recommended. The review follows the Interim Guidance from the Cochrane Rapid Reviews Methods Group [17]. The protocol of this study was registered on the Open Science Framework (DOI https://doi.org/10.17605/OSF.IO/XS687). Details about the search string, eligible criteria, and data extraction are available in the protocol.

### 2.1. Definition of Vaccination Type

For the purpose of the study, a mandatory vaccination is defined as a vaccination that every child must receive by law without the possibility for the parent to choose to accept whether they take it or not. A recommended vaccination is a vaccination that is included in the national immunization program for all or for some specific groups independent of being funded or not [18].

### 2.2. Included Vaccinations

We included all vaccinations provided to individuals aged 0–16 years as listed on the European Centre for Disease Prevention and Control (ECDC) website [19], specifically: mumps; measles; rubella; varicella; tetanus-diphtheria-pertussis; Haemophilus influenza type B; hepatitis B; poliomyelitis; meningococcal disease; and pneumococcal disease. Routine vaccinations for adults, travelers, and healthcare professionals were excluded from this analysis.

### 2.3. Sources

We searched national pediatric immunization schedules on PubMed, Google and the ECDC website, as of September 2024. Furthermore, we identified relevant documents regarding the introduction of mandatory pediatric vaccination policies and laws, published between January 2014 and September 2024. We also conducted a backward citation screening of all literature reviews on the topic to identify and gather additional relevant documents. The year 2014 was considered a reference point as it marks the Regional European Office of the WHO’s recognition of the critical need to intensify efforts in promoting pediatric vaccinations [20].

### 2.4. Data Collection and Analysis

After conducting a double-blind screening of the records, we identified the countries that had introduced any mandatory pediatric vaccinations and extracted data on the specific vaccines and their corresponding year of introduction. Subsequently, we conducted a descriptive analysis and summarized the results in tables. Results are presented by antigen and by country.

## 3. Results

### 3.1. Childhood Vaccination Schedules in EU/EEA in 2024

As shown in Table 1, among the 30 EU/EEA countries, five countries (Austria, Greece, Liechtenstein, Luxembourg, and Spain) recommend all vaccinations included in their national immunization schedules. Belgium has also adopted a comprehensive vaccination policy, recommending all vaccines except for poliomyelitis vaccination, which is mandatory. Ten countries (Bulgaria, Croatia, the Czech Republic, France, Hungary, Italy, Latvia, Poland, Slovakia, and Slovenia) have established mandatory vaccination policies for the following vaccines: diphtheria; tetanus; pertussis; hepatitis B; Haemophilus influenzae type b (Hib); poliomyelitis; measles; mumps; and rubella. Notably, in Austria, Belgium, Cyprus, and the Czech Republic, varicella vaccination is recommended but not covered by the national healthcare system (NHS), whereas only in Austria and Poland is the meningococcal vaccine recommended but not funded by the NHS.

#### 3.1.1. Mumps, Measles, Rubella

Vaccination schedules for mumps, measles, and rubella (MMR) show numerous variations across the 30 countries analyzed. These vaccines are mandatory in 10 countries, including Bulgaria, Croatia, the Czech Republic, France, Hungary, Italy, Latvia, Poland, Slovakia, and Slovenia. In Germany, measles vaccination is mandatory, whereas mumps and rubella vaccinations are recommended. In all the remaining countries, the MMR vaccination is recommended but not mandatory (Table 1) (Figure 1).

#### 3.1.2. Varicella

Varicella vaccination exhibits notable differences in policy. As of 2024, we found that three countries (Italy, Latvia, and Hungary) have made varicella vaccination mandatory. In contrast, 12 countries recommend the vaccine, while 15 countries have yet to include it in their national recommendations (Table 1) (Figure 2).

#### 3.1.3. Tetanus-Diphtheria-Pertussis

For tetanus-diphtheria-pertussis (Tdap), all three vaccines are mandatory in 10 countries (Bulgaria, Croatia, the Czech Republic, France, Hungary, Italy, Latvia, Malta, Poland, Slovakia, and Slovenia), while in Malta, only tetanus and diphtheria are required. In the remaining 19 countries, Tdap vaccines are recommended but not obligatory (Table 1) (Figure 3).

#### 3.1.4. Haemophilus Influenza Type B

Vaccination against Haemophilus influenzae type B (Hib) follows a similar pattern. Hib vaccination is mandatory in 10 countries (Bulgaria, Croatia, the Czech Republic, France, Hungary, Italy, Latvia, Poland, Slovakia, Slovenia), while it remains recommended in the other 20 (Table 1) (Figure 4).

#### 3.1.5. Hepatitis B

Regarding hepatitis B, the vaccine is mandatory in 10 countries (Bulgaria, Croatia, the Czech Republic, France, Hungary, Italy, Latvia, Poland, Slovakia, and Slovenia) and in the remaining countries it is recommended, particularly for newborns of hepatitis B-positive mothers to prevent transmission. The only country where vaccination is neither mandated nor recommended is Iceland (Table 1) (Figure 5).

#### 3.1.6. Poliomyelitis

For poliomyelitis, vaccination is mandatory in 12 countries (Belgium, Bulgaria, Croatia, the Czech Republic, France, Hungary, Italy, Latvia, Malta, Poland, Slovakia, and Slovenia) and recommended in the other 18 (Table 1) (Figure 6).

#### 3.1.7. Meningococcal Disease

In the case of meningococcal disease, the vaccine is recommended in 19 countries, while France has implemented mandatory vaccination. Notably, 10 countries (Bulgaria, Croatia, Denmark, Estonia, Finland, Latvia, Norway, Romania, Slovenia, and Sweden) have not yet included a recommendation for the meningococcal vaccine in their national immunization programs (Table 1) (Figure 7).

#### 3.1.8. Pneumococcal Disease

In 22 countries, vaccination against pneumococcal disease is recommended as part of their national immunization program. However, in seven countries (Bulgaria, Croatia, France, Hungary, Latvia, Poland, and Slovakia), the vaccine is mandatory. Estonia is the only country where no recommendation for pneumococcal vaccination has been implemented (Table 1) (Figure 8).

### 3.2. Childhood Vaccination Policies in EU/EAA in 2014–2024

Among 814 records, we included 13 documents on the introduction of childhood mandatory vaccinations, as reported in the PRISMA flow diagram (Figure 9). We identified six EU/EAA countries that implemented at least one mandatory pediatric vaccination between 2014 and 2024, based on the results of gray and scientific literature (Table 2). In Italy, the 2017 reform expanded the list of compulsory vaccines from four to ten, to include diphtheria, tetanus, pertussis, hepatitis B, poliovirus, Haemophilus influenzae type b, measles, mumps, rubella, and varicella. Similarly, France introduced new mandatory vaccination requirements in 2018, increasing the number from three to eleven by adding immunizations for measles, mumps, rubella, hepatitis B, Haemophilus influenzae type b, pneumococcus, and meningococcus C. However, starting in 2021, the meningococcal vaccination returned to being only recommended. Poland introduced mandatory pneumococcal vaccination in 2017, while Croatia added this requirement in 2019. Hungary also introduced mandatory varicella vaccination in 2019. In 2020, Germany implemented mandatory measles vaccination for all children attending school or daycare, along with healthcare workers.

## 4. Discussion

The findings from this review highlight the diverse landscape of pediatric vaccination policies across the EU and EEA. In particular, we found that 13 countries had at least one mandatory pediatric vaccination, while 17 countries had no mandatory vaccinations, relying only on recommendations. These data show great variability at the EU/EEA level, which is also found at the global level, with some countries having introduced forms of obligation (such as the US) while others have not (such as the UK, Japan, and Australia) [34].

Mandatory childhood vaccinations have proven to be a highly effective tool in improving vaccination coverage and reducing the incidence of VPDs [23,35,36,37,38]. Studies show that countries with mandatory vaccination policies generally achieve higher coverage rates compared to those relying solely on recommendations. For instance, mandatory vaccination has been associated with a 3.71% increase in measles vaccination and a 2.14% rise in pertussis vaccination in countries with mandatory laws when compared with countries that did not improve mandatory vaccination. Moreover, it has been observed that mandatory laws could reduce the incidence of VPDs, such as measles [39]. Despite this clear evidence, less than half of the 30 analyzed countries have implemented at least one mandatory pediatric vaccination [39]. In addition, several studies have evaluated vaccination coverage trends following the introduction of the compulsory law. For example, an increase in coverage was observed in Italy after the 2017 law, with measles coverage at 30 months of age rising from 87.3% in 2016 to 91.8% in 2017 and 94.1% as of June 2018 [35]. Similarly, an increase in vaccination coverage was observed in France after the introduction of the mandatory law, with hexavalent coverage increasing from 93.1% in 2017 to 98.6% in 2018, and a 36.4% increase for meningococcal C coverage was observed in the same period [23]. In Germany, a drastic decline in measles cases has been observed following the introduction of the compulsory law, from about 500–1000 per year to 10–80 per year since 2020 [40]. In this sense, the compulsory law may have helped reduce the spread of measles, although vaccination coverage increased only slightly comparing the periods before and after the introduction of the new law [41].

The selection of which vaccines to mandate typically reflects the severity and transmission potential of the diseases they prevent. Vaccines for highly contagious and severe diseases, such as measles and poliomyelitis, are often prioritized in mandatory vaccination programs due to their potential to cause widespread outbreaks and severe health consequences. For example, 11 EU/EEA countries have made the MMR vaccine mandatory, responding to periodic measles outbreaks in Europe. As highlighted by the World Health Organization and the ECDC [42,43], these outbreaks are mainly related to low vaccination coverage. The disruption of primary care services that occurred during the COVID-19 pandemic also contributed to the decrease in coverage. Similarly, the reduction in vaccination coverage observed during the COVID-19 pandemic resulted in a resurgence of VPDs, as in the case of pertussis [44]. As such, it is critical to close the immunization gaps and achieve and maintain high vaccination coverage through a rapid, coordinated, and effective response [45].

Vaccines for diseases with lower transmission rates or less severe outcomes, like varicella, are more often recommended rather than mandated. Only a few countries, including Italy and Hungary, have made varicella vaccination compulsory, while most others opt for recommendations based on risk factors and individual health profiles.

The structure of healthcare systems may play a crucial role in the effectiveness of mandatory vaccination policies. Countries with centralized systems, such as Italy and France, have demonstrated greater success in implementing these policies due to their streamlined decision-making processes and cohesive management structures. This centralization allows for uniform enforcement and higher overall vaccination rates. Italy’s introduction of mandatory vaccinations in 2017 and France’s expansion of its program in 2018 are key examples of how these systems can quickly boost coverage. In contrast, countries with decentralized systems, like Germany and Netherlands, face more challenges. The fragmentation of healthcare delivery and a greater emphasis on individual autonomy make it difficult to uniformly enforce mandatory vaccinations, often leading these countries to rely on recommendations instead, resulting in regional disparities in coverage [16,46,47].

Cultural attitudes also deeply influence the acceptance of vaccination policies. For example, in Eastern European countries, where strong state involvement in public health is more common and accepted, mandatory vaccinations are met with less resistance. Contrarily, in countries where personal freedom is highly valued, there is greater opposition to mandatory vaccination policies, and a preference is often given to voluntary or recommended vaccinations. Imposing mandates in these contexts can sometimes exacerbate vaccine hesitancy, especially when not accompanied by effective public education campaigns that address concerns around autonomy and state intervention [48].

Despite the clear benefits of mandatory vaccination, such policies raise important ethical, legal, and social considerations. The balance between individual autonomy and public health needs is a key challenge, especially in countries that prioritize personal freedom. In such contexts, mandatory vaccination policies can lead to resistance and even legal challenges, particularly when nonmedical exemptions are not allowed. Notably, countries without nonmedical exemptions have seen a more pronounced decrease in incidence of VPDs, particularly measles, when mandatory vaccination is enforced. Therefore, to ensure compliance and minimize public resistance, it is crucial to complement mandates with public education and communication strategies that build trust and address concerns about personal rights [49,50]. Interestingly, considering the impact of vaccination policies on public uptake, some countries achieve high vaccination coverage without mandates. For instance, different countries maintain high vaccination rates through official recommendation programs, driven by strong public trust in the healthcare system and effective communication and public engagement strategies [46,51]. This suggests that while mandatory policies can be effective in increasing vaccination coverage, they are not the only strategies to achieve high uptakes. Building public trust and ensuring easy access to vaccines are equally important, and, in some cases, may be sufficient to meet immunization goals.

The COVID-19 pandemic further underscored the importance of flexible and resilient vaccination strategies. The pandemic disrupted routine immunization programs and heightened vaccine hesitancy, particularly regarding the rapid development of COVID-19 vaccines. Countries had to quickly adjust their policies to maintain coverage for routine immunizations while addressing emerging public health threats. This experience highlighted the need for adaptable vaccination systems capable of responding swiftly to both existing and new challenges and emphasized the critical role of effective public communication in reinforcing the importance of vaccines during times of crisis [52,53].

Despite the comprehensive scope of this review, it has certain limitations. The rapid review methodology, while efficient, may have led to the exclusion of relevant documents. Additionally, variability in national data collection and reporting practices could affect the comparability of results across countries. Finally, this study focuses primarily on policy analysis, leaving the long-term health outcomes and the effectiveness of these policies over time open for further investigation.

## 5. Conclusions

This review reveals the complex and evolving picture of pediatric vaccination policies across Europe. Countries have adopted a wide range of strategies to combat declining vaccination rates and the resurgence of vaccine-preventable diseases, from expanding mandatory vaccination programs to focusing on recommendation and public education. The variation in policies underscores the importance of context in shaping public health interventions. In this sense, the ongoing efforts to harmonize vaccination policies and address public concerns will be key to ensuring sustained progress in preventing infectious diseases across Europe.

Countries with successful vaccination programs, whether through mandates or recommendations, share common features such as effective healthcare systems, accessible vaccination services, and strong public trust in health authorities. As new vaccines are introduced and vaccine hesitancy continues to pose challenges, European countries will need to adapt their strategies to maintain or increase vaccination coverage. Future research should focus on the long-term outcomes of these different approaches, particularly considering the impact of COVID-19 on public attitudes toward vaccination.

## Figures and Tables

**Figure 1 vaccines-12-01296-f001:**
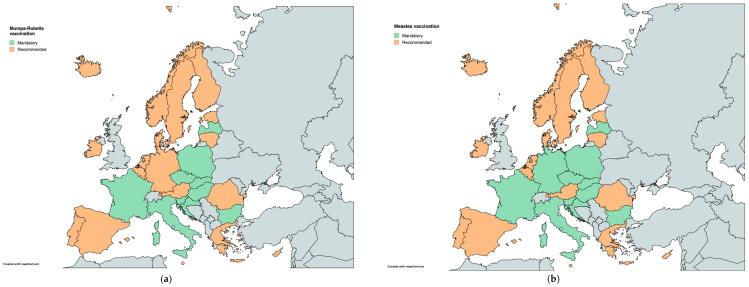
Childhood vaccination policies in EU/EEA countries in 2024 for (**a**) mumps-rubella and (**b**) measles.

**Figure 2 vaccines-12-01296-f002:**
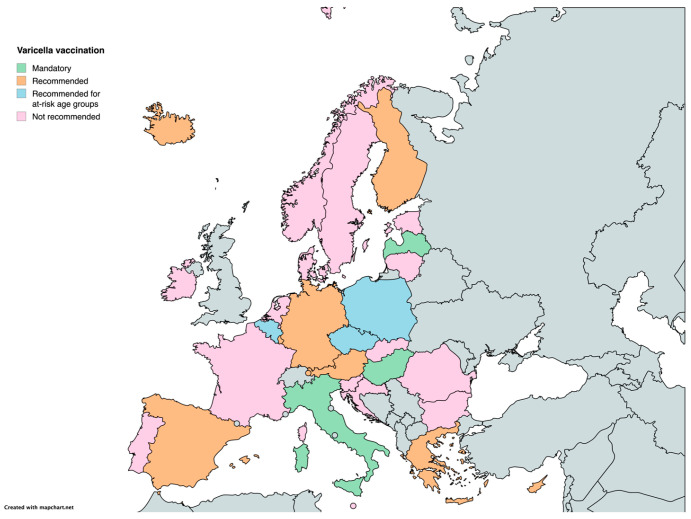
Childhood vaccination policies in EU/EEA countries in 2024 for varicella.

**Figure 3 vaccines-12-01296-f003:**
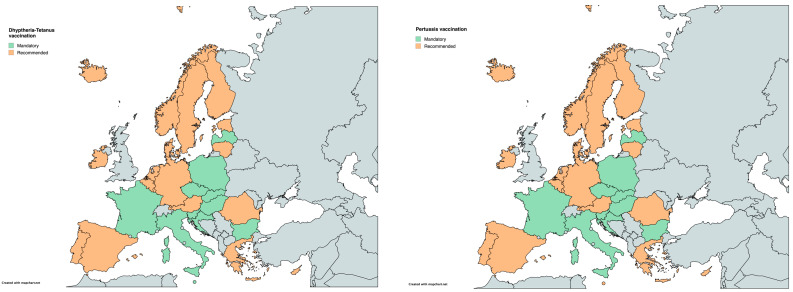
Childhood vaccination policies in EU/EEA countries in 2024 for (**a**) tetanus-diphtheria and (**b**) pertussis.

**Figure 4 vaccines-12-01296-f004:**
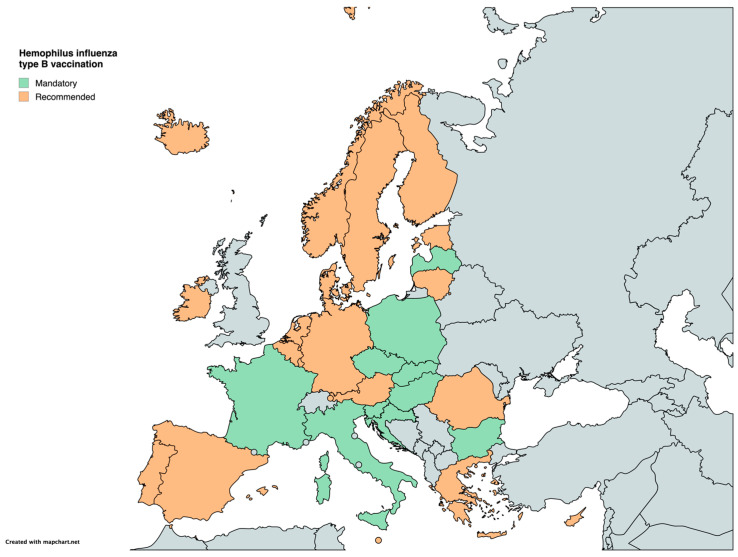
Childhood vaccination policies in EU/EEA countries in 2024 for Haemophilus influenza type B.

**Figure 5 vaccines-12-01296-f005:**
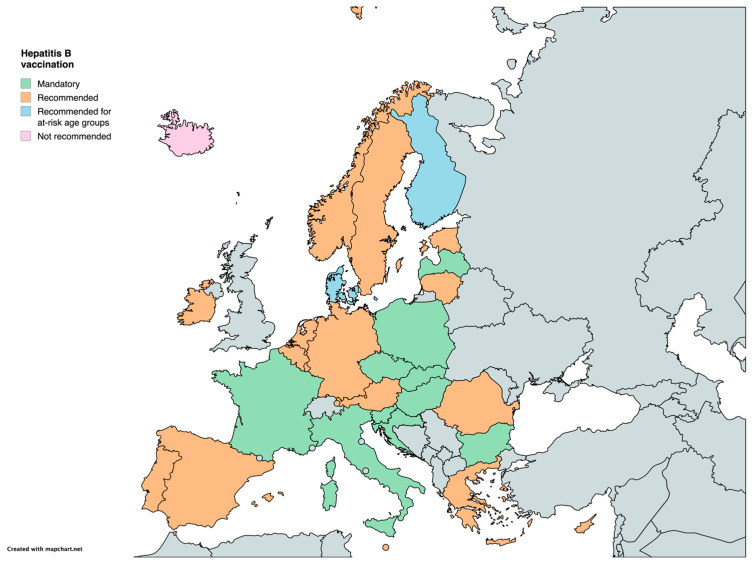
Childhood vaccination policies in EU/EEA countries in 2024 for Hepatitis B.

**Figure 6 vaccines-12-01296-f006:**
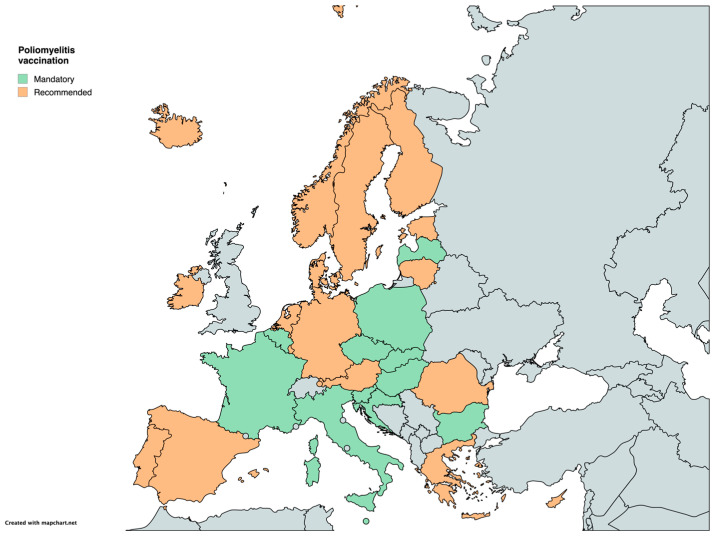
Childhood vaccination policies in EU/EEA countries in 2024 for Poliomyelitis.

**Figure 7 vaccines-12-01296-f007:**
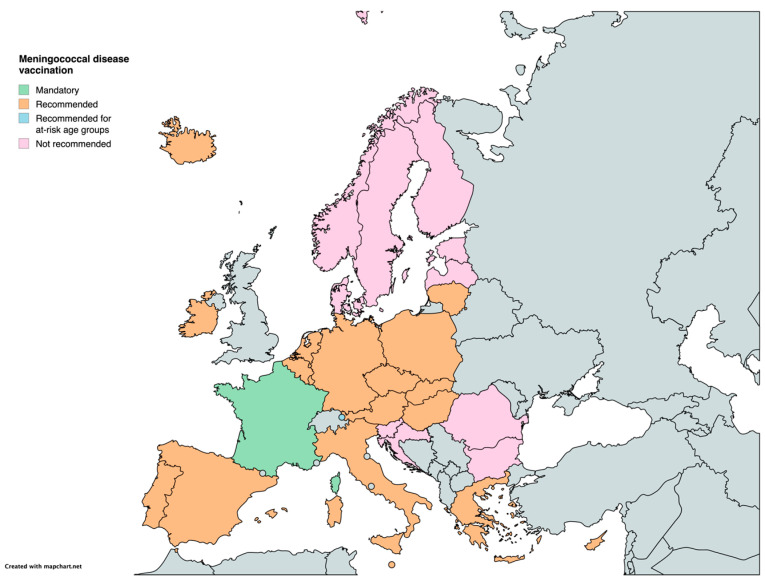
Childhood vaccination policies in EU/EEA countries in 2024 for meningococcal disease.

**Figure 8 vaccines-12-01296-f008:**
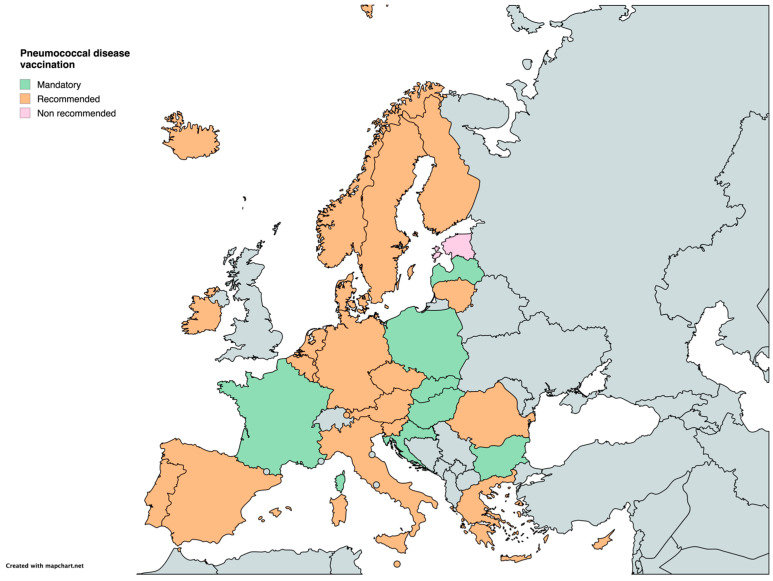
Childhood vaccination policies in EU/EEA countries in 2024 for Pneumococcal disease.

**Figure 9 vaccines-12-01296-f009:**
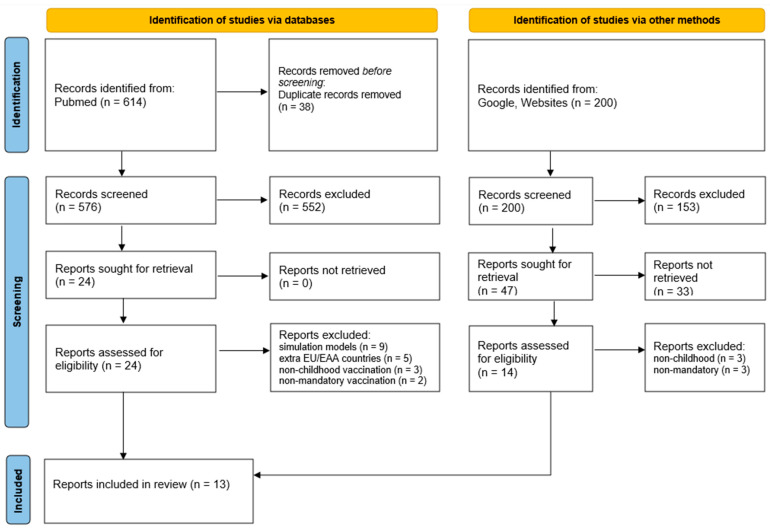
PRISMA flow chart for records selection.

**Table 1 vaccines-12-01296-t001:** Childhood vaccination schedules in EU/EAA countries in 2024.

Country	Diphtheria	Tetanus	Pertussis	Hepatitis B	Hib	Poliomyelitis	Measles	Mumps	Rubella	Varicella	Meningococcal Disease	Pneumococcal Disease
Austria											*	*	
Belgium											*	**	
Bulgaria												
Croatia													
Cyprus											*		***	
Czech Republic											*		
Denmark												
Estonia													
Finland												
France														****	
Germany													
Greece														
Hungary													
Iceland												
Ireland													
Italy													
Latvia													
Liechtenstein													
Lithuania												
Luxembourg																		
Malta													
Netherlands													
Norway													
Poland											*	
Portugal												
Romania												
Slovakia													
Slovenia													
Spain													
Sweden													


: mandatory; 

: recommended; 

: not recommended; 

: recommended for at-risk age groups; 

: mandatory for specific group only * not funded by the NHS; ** from the 14th to 16th years not funded by the NHS; *** from the 2nd to 16th years not funded by the NHS; **** recommended at the 3rd month.

**Table 2 vaccines-12-01296-t002:** Mandatory childhood vaccination policies in EU/EAA countries from 2014 to 2024.

Country	Vaccination	Year of Policy/Law Introduction	References
Croatia	Pneumococcal disease	2019	[21,22]
France	Diphtheria, tetanus, poliomyelitis, pertussis, Hib, hepatitis B, pneumococcal disease, Meningococcal disease, measles, mumps, rubella	2018	[23,24,25]
Germany	Measles	2020	[26,27]
Hungary	Varicella	2019	[28]
Italy	Diphtheria, tetanus, poliomyelitis, pertussis, Hib, hepatitis B, measles, mumps, rubella, varicella	2017	[29,30,31,32]
Poland	Pneumococcal disease	2017	[33]

## Data Availability

Data were collected from publicly available international databases and websites.

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
