# Peer review of "Childhood Mandatory Vaccinations: Current Situation in European Countries and Changes Occurred from 2014 to 2024"

_vaccines, 2024, doi:10.3390/vaccines12111296_

Round 1

Reviewer 1 Report

Comments and Suggestions for Authors

This review summaized the vaccination recommendation in EU.

I have few comments to the authors:

1) While UK and Switzerland are not part of EU due to politicial reasons, as they are close to EU countries, I would suggest to add the data from these countries in a separate paragraph if possible

2) It will be good if we can also compare the data in other major places in the World like USA, Canada, Japan etc.

3) Any changes to these national recommendation, esp. 2024?

4) Fig 9 should appear earlier

5) Conclusion too long, please shorten

Author Response

This review summaized the vaccination recommendation in EU.

We thank the reviewer for all the valuable comments that helped us improve the manuscript to make it suitable for publication.

I have few comments to the authors:

1) While UK and Switzerland are not part of EU due to political reasons, as they are close to EU countries, I would suggest to add the data from these countries in a separate paragraph if possible

The purpose of the study is to map the mandatory vaccination landscape among EU/EEA countries. Our research question as well as the methodology used of consulting the ECDC website and gray literature is based on the inclusion of these countries, thus excluding all non-EU/EEA countries. Therefore, it is beyond the scope of this paper to include UK and Switzerland as well. Moreover, data about these two countries are not available on the ECDC website. However, we added some insights about UK and Switzerland in the discussions.

2) It will be good if we can also compare the data in other major places in the World like USA, Canada, Japan etc.

We added some insights about these countries in the discussion section.

3) Any changes to these national recommendation, esp. 2024?

We analyse all the changes occurred in the last 10 years in table 2.

4) Fig 9 should appear earlier

Figure 9 is positioned within the paragraph “Childhood vaccination policies in EU/EEA in 2014-2024.” We believe that it is more important to focus on the current sitation and, therefore, the first paragraph of the results is the one on Childhood vaccination schedules in EU/EEA in 2024. We believe it is important to maintain this logical order and would therefore be of the opinion to keep the figure within paragraph 3.2.

5) Conclusion too long, please shorten

We believe that the final key messages in the conclusion section are important, and we would keep them.

Reviewer 2 Report

Comments and Suggestions for Authors

Dear authors

Thank you for the opportunity to review the manuscript entitled “Childhood mandatory vaccinations: history and current situation in European Countries in the last ten years”. For probably publication on vaccines Journal.

My decision is Accept after major revisions.

The mandatory of vaccination has been suggested as a strategy to increase vaccination uptake but after the Covid -19 pandemic several studies showed that among vaccine skeptics, the reduction of confidence in the government's management include the mandatory guidelines for vaccination.

The paper is interesting and deal with the emerging topic. The introduction is clear and well arranged. The methodology sounds good. The discussion is good even could be improved.

The main question for the authors is what new brings the present manuscript. Similar data about mandatory vaccines we can find on ECDC official page for each country of Europe: https://vaccine-schedule.ecdc.europa.eu/ ,

However, I suggest to authors the follow corrections to improve their manuscript.

·        Reform the title to: “Childhood mandatory vaccinations: current situation in European Countries”.

·        Lines 148-149 “Varicella vaccination exhibits notable differences in policy. As of 2024, three countries (Italy, Latvia, and Hungary) have made varicella vaccination mandatory”: please add the source for each country for this decision.

·        Lines 236-237: enhance your position with reference.

·        Add and compare the vaccination coverage for the vaccines before and after the mandatory.

·        Table 2: exam the association between the vaccination rates and mandatory vaccination policies in countries of interest (e.g. measles for Deutschland). What the deference’s to vaccination coverage between 2020-2024.

·        Another important point for discussion is the recent epidemic of pertussis in Europe and how this connected with low vaccination coverage and mandatory policies of vaccination.

·        Line 247-248: add the explanation of periodic outbreaks (what is the reason?) the low vaccination coverage the missing shots of the booster doses?

·        Lines 257-264: Could you discuss your position with the position of Stefano Crenna et al “In Italy, regulations about vaccinations are controversial and, to some extent, inconsistent. Even though the childhood vaccinations are mandatory by law (Italian Law n. 891/1939, n. 292/1963, n.51/1966 and n. 165/1991), the limited deterrent effectiveness of the sanctioning system, and the changes introduced by the Italian Constitutional Law n. 3/2001 (devolution of almost all the competences and responsibilities in health matters to the Regions and the Autonomous Provinces), were the fertile ground in which new vaccine policies were generated and developed, radically different from the existing ones: many Regions, based on what was decided in 2005 - on an experimental basis - by the State-Regions Conference, decided to abolish the vaccination obligation and/or to stop the imposition of administrative sanctions on non-compliant parents ( doi: 10.4081/jphr.2018.1523) .

·        The references should be reform according to Journal suggestions

Author Response

The mandatory of vaccination has been suggested as a strategy to increase vaccination uptake but after the Covid -19 pandemic several studies showed that among vaccine skeptics, the reduction of confidence in the government's management include the mandatory guidelines for vaccination. The paper is interesting and deal with the emerging topic. The introduction is clear and well arranged. The methodology sounds good. The discussion is good even could be improved. The main question for the authors is what new brings the present manuscript. Similar data about mandatory vaccines we can find on ECDC official page for each country of Europe: https://vaccine-schedule.ecdc.europa.eu/. However, I suggest to authors the follow corrections to improve their manuscript.

We thank the reviewer for all the valuable comments that helped us improve the manuscript to make it suitable for publication.

  • Reform the title to: “Childhood mandatory vaccinations: current situation in European Countries”.

We changed the title in “Childhood mandatory vaccinations: current situation in European Countries and changes occurred from 2014 to 2024”. We believe it is important to keep in the title the 10 years’ analysis that we conducted, as it is a key part of the paper.

  • Lines 148-149 “Varicella vaccination exhibits notable differences in policy. As of 2024, three countries (Italy, Latvia, and Hungary) have made varicella vaccination mandatory”:please add the source for each country for this decision.

We specified that this statement comes from the results of our research by changing the sentence to: “Varicella vaccination exhibits notable differences in policy. As of 2024, we found that three countries (Italy, Latvia, and Hungary) have made varicella vaccination mandatory.” We added the reference of the ECDC scheduler in the methods section.

  • Lines 236-237: enhance your position with reference.

We updated the references.

  • Add and compare the vaccination coverage for the vaccines before and after the mandatory.

The purpose of the study is to map the mandatory vaccination landscape among EU/EEA countries. Evaluating the impact of the law is a key topic that, however, is beyond the scope of this paper. In any case, we have added in the initial section of the discussion this part. We reported the main findings related to the implementation of the mandatory laws in EU/EEA countries, such as Italy and France. These are two examples of increased coverage following the introduction of the law, reported by several studies both at the national level and by analyzing regional contexts. All the references have been added.

  • Table 2: exam the association between the vaccination rates and mandatory vaccination policies in countries of interest (e.g. measles for Deutschland). What the deference’s to vaccination coverage between 2020-2024.

As mentioned before, the purpose of the study is to map the mandatory vaccination landscape among EU/EEA countries and evaluating the impact of the law is beyond the scope of this paper. We did not find paper assessing the impact of mandatory law in Germany. However, we reported data according to coverages in the country, assessing the trend. The reference has been added.

  • Another important point for discussion is the recent epidemic of pertussis in Europe and how this connected with low vaccination coverage and mandatory policies of vaccination.

 We added our considerations about the pertussis outbreaks in EU, especially because of the COVID-19 pandemic. Appropriate references have been added.

  • Line 247-248: add the explanation of periodic outbreaks (what is the reason?) the low vaccination coverage the missing shots of the booster doses?

 We added the reasons of periodic outbreaks, with a specific focus on the effect of the COVID-19 pandemic. Appropriate references have been added.

  • Lines 257-264: Could you discuss your position with the position of Stefano Crenna et al “In Italy, regulations about vaccinations are controversial and, to some extent, inconsistent. Even though the childhood vaccinations are mandatory by law (Italian Law n. 891/1939, n. 292/1963, n.51/1966 and n. 165/1991), the limited deterrent effectiveness of the sanctioning system, and the changes introduced by the Italian Constitutional Law n. 3/2001 (devolution of almost all the competences and responsibilities in health matters to the Regions and the Autonomous Provinces), were the fertile ground in which new vaccine policies were generated and developed, radically different from the existing ones: many Regions, based on what was decided in 2005 - on an experimental basis - by the State-Regions Conference, decided to abolish the vaccination obligation and/or to stop the imposition of administrative sanctions on non-compliant parents ( doi: 10.4081/jphr.2018.1523) .

The National Health Service in Italy is highly decentralized. However, the Central government sets the national benefits package and allocates funding for the regional health systems, including vaccinations. This means that while regions may have discretion of service delivery, some points and obligations are set at the national level. This is the case, for example, with the 2017 mandatory vaccination law. This law was approved at the central level, indicating which vaccines should be made mandatory and which recommended. Similarly, the Immunization Plan (Piano Nazionale Prevenzione Vaccinale – PNPV) and the “calendar for life” are updated centrally by providing implementation guidance for the regions, which may, however, deliver services differently (this is the case of delivery for specific age groups or free of charge for certain vaccinations). In any case, regions must ensure the minimum standard set at central level.

The references should be reform according to Journal suggestions

Ok thank you

Round 2

Reviewer 2 Report

Comments and Suggestions for Authors

Thank you for the revised form